Measuring coral calcification under ocean acidification: methodological considerations for the 45Ca-uptake and total alkalinity anomaly technique

Cohen Stephanie stephanie.cohen@epfl.ch 1 2 3
Krueger Thomas 3
Fine Maoz 1 2
1 The Interuniversity Institute for Marine Science , Eilat , Israel
2 The Mina & Everard Goodman Faculty of Life Sciences, Bar-Ilan University , Ramat Gan , Israel
3 Laboratory for Biological Geochemistry, School of Architecture, Civil and Environmental Engineering, École Polytechnique Federale de Lausanne , Lausanne , Switzerland
Ragazzola Federica
Electronic publication date: 2017 Sep 1
Publication date: 2017
Volume: 5
Electronic Location ID: e3749
Received 2017 May 24; Accepted 2017 Aug 8
Copyright: ©2017 Cohen et al.
Copyright year: 2017
Copyright holder: Cohen et al.
License: This is an open access article distributed under the terms of the Creative Commons Attribution License, which permits unrestricted use, distribution, reproduction and adaptation in any medium and for any purpose provided that it is properly attributed. For attribution, the original author(s), title, publication source (PeerJ) and either DOI or URL of the article must be cited.
License URL: https://creativecommons.org/licenses/by/4.0/

Keywords: Gross calcification, Net calcification, Coral dissolution, Biomineralisation, Coral reefs

Funding: Israel Science Foundation This study was funded by an Israel Science Foundation grant to M Fine. There was no additional external funding received for this study. The funders had no role in study design, data collection and analysis, decision to publish, or preparation of the manuscript.

==============================
As the oceans become less alkaline due to rising CO2 levels, deleterious consequences are expected for calcifying corals. Predicting how coral calcification will be affected by on-going ocean acidification (OA) requires an accurate assessment of CaCO3 deposition and an understanding of the relative importance that decreasing calcification and/or increasing dissolution play for the overall calcification budget of individual corals. Here, we assessed the compatibility of the 45Ca-uptake and total alkalinity (TA) anomaly techniques as measures of gross and net calcification (GC, NC), respectively, to determine coral calcification at pHT 8.1 and 7.5. Considering the differing buffering capacity of seawater at both pH values, we were also interested in how strongly coral calcification alters the seawater carbonate chemistry under prolonged incubation in sealed chambers, potentially interfering with physiological functioning. Our data indicate that NC estimates by TA are erroneously ∼5% and ∼21% higher than GC estimates from 45Ca for ambient and reduced pH, respectively. Considering also previous data, we show that the consistent discrepancy between both techniques across studies is not constant, but largely depends on the absolute value of CaCO3 deposition. Deriving rates of coral dissolution from the difference between NC and GC was not possible and we advocate a more direct approach for the future by simultaneously measuring skeletal calcium influx and efflux. Substantial changes in carbonate system parameters for incubation times beyond two hours in our experiment demonstrate the necessity to test and optimize experimental incubation setups when measuring coral calcification in closed systems, especially under OA conditions.

Introduction

Continual increases in atmospheric CO2-concentration has led to measurable changes in the carbonate chemistry of the oceanic system, summarized under the term ocean acidification (OA; Kleypas et al., 2006). These changes involve an increase in total dissolved inorganic carbon (DIC) and a shift in the carbon equilibrium towards CO2, which lead to a reduction in overall aragonite saturation state (Ωarag) and pH (Kleypas et al., 1999; Zeebe & Wolf-Gladrow, 2001). Current worst case climate models project a further decrease of surface seawater pH by 0.3 − 0.4 pH units until the end of the 21st century (IPCC, 2013). These continuing shifts in seawater pH and aragonite saturation state affect many marine organisms that form biogenic aragonite; the modern day form of calcium carbonate in scleractinian reef-building corals (Anthony et al., 2008; McCulloch et al., 2012; Orr et al., 2005; Ries, Cohen & McCorkle, 2009).

Coral calcification is the fundamental biological process that provides the physical three-dimensional platform for the existence of coral reef communities. However, deposited skeleton in a reef is subject to various forms of physical, chemical and biologically-mediated erosion, which causes the dissolution of reef sediments and skeleton. This dissolution is part of the natural turnover of matter in the reef community and can offset 20–30% of reef calcification (Barnes, 1988; Silverman, Lazar & Erez, 2007). It is widely accepted that OA will increase overall calcium carbonate dissolution of coral reef communities (e.g., Andersson, Mackenzie & Gattuso, 2011; Eyre, Andersson & Cyronak, 2014; Silverman et al., 2009). With scleractinian corals as major reef builders, a large body of research has thus been dedicated to the performance of individual coral species under varying degrees of OA. While studies generally demonstrate a decline in net calcification (NC) to varying degrees, a few show also insensitivity of some species to OA conditions (reviewed in Table S1; Chan & Connolly, 2013). The variety in the calcification-dependent growth response to low seawater pH may reflect the true variability/plasticity in the ability of some corals to maintain calcification rates over a broad range of seawater Ωarag (McCulloch et al., 2012). On the other hand, there is also a high degree of methodological variation in design, choice of pH lowering agents, interaction of one or multiple environmental factors along with pH (e.g., temperature, light), animal acclimation period, experimental incubation period, and employed techniques to determine calcification rates that contribute to the observed variability. Recognizing the fundamental impact that OA has on calcifying organisms, there is now a renewed interest for calcification studies of corals.

While the occurrence of net dissolution under OA conditions and during the night has been observed for some species (cf. Table S1), the contribution of skeleton dissolution (gross dissolution; GD) to changes in NC remains unknown. Commonly, it is assumed that changes of NC can be the result of a decrease in gross calcification (GC, the calcification rate before any deductions for dissolution), increased GD, or a combination of both following the equation NC = GC-GD (Fig. 1; Andersson & Mackenzie, 2012; Comeau et al., 2014; Langdon, Gattuso & Andersson, 2010). However, this conceptual relationship has never been experimentally validated to see to what extent it is possible to quantify dissolution from measurements of NC and GC.

Figure 1 Conceptual framework of calcification in isolated coral colonies.

The specific calcification rate that determines the transition of calcium carbonate from its source to its sink is affected by the opposing effects of the biologically determined gross calcification rate and the dissolution rate. Depending on their respective magnitudes, the coral experiences net calcification or net dissolution of skeletal material. Quantifying net calcification in corals is commonly achieved by monitoring the change in source (via forms of titration) or sink (e.g., buoyant weight) material. When assessed over short time periods, where gross dissolution is negligible/not detectable, sensitive methods that can detect incorporated labels provide an estimate for gross calcification. Note that changes in skeletal density and porosity under low pH might alter gross calcification without involving processes of dissolution.

The most common methods that are widely used to measure NC in living corals are the “buoyant weight” and “total alkalinity” technique (Table 1; Langdon, Gattuso & Andersson, 2010). While the total alkalinity (TA) anomaly technique is based on tracking changes in TA in the incubation medium (Chisholm & Gattuso, 1991; Smith & Key, 1975), the buoyant weight (BW) technique is a direct measure of colony growth via repeated determinations of overall weight (Jokiel & Maragos, 1978). The TA is measured by acidimetric titration of the total inorganic carbon in the seawater sample. From the TA equation one can infer the calcification and dissolution of CaCO3: precipitation of 1 mole CaCO3 leads to a 2 moles decrease in TA, whereas dissolution causes the reverse effect (Wolf-Gladrow et al., 2007). While these two methods are generally considered to give estimates of NC and provide comparable calcification rates (Schoepf et al., 2017), the only method likely to provide measurements of gross calcification (GC) over short incubation periods is the 45Ca-labelling technique. GC is commonly determined by direct incorporation of the radio-isotope 45Ca into the skeleton using fully covered microcolonies (Goreau, 1959; Goreau & Goreau, 1960; Tambutté et al., 1995). The procedure involves incubation in 45Ca-labeled seawater, dissolving the skeleton in acid and measuring the incorporated radioactivity with a liquid scintillation counter. The TA and 45Ca techniques are the only ones considered to have sufficient resolution to detect small changes in calcification (Chisholm & Gattuso, 1991; Langdon, Gattuso & Andersson, 2010) and to our knowledge, Smith and Roth (in Smith & Kinsey, 1978) and Tambutté et al. (1995) are the only studies to date to provide a direct comparison of the TA and 45Ca techniques for corals under ambient pH conditions. The accurateness and compatibility of both methods for incubations under reduced pH conditions has not yet been experimentally tested.

Table 1 Methods for measuring calcification in tropical corals.

Overview of the most common methods and their employed acclimation and incubation times for measuring OA effects on calcification in tropical hermatypic corals. The detailed summary for each species and its approximate change in calcification is provided in Table S1. The number of studies using the mentioned acclimation and incubation time periods is given in brackets. Note that this number might not add up to the given total number, since some studies might have used different acclimation times when performing multiple experiments or involving different species. BW, buoyant weight; TA, total alkalinity.

Method	No. of species	No. of studies	Acclimation timea	Incubation time	
BW	32	26	<1 month (10)	–	
			1–2 month (13)	–	
			3–12 month (5)	–	
			>1 year (2)	–	
TA	17	13	No acclimation (5; 2b)	<2.5 h (7)	
			Several hours (2)	3–7 h (3)	
			<1 month (4)	>12 h (2)	
			1–2 months (2)		
Ca isotopes	1	3	No acclimation (1)	3–7 h (2; 45Ca)	
			1–2 months (2 )	>12 h (1; 43Cac)	
Other methodsc	3	5	No acclimation (2; 1b)	1–2 months (2)	
			>1 year (3)	>1 year (1)	
Notes.

a Time required for acclimation and incubation time were summed up for coral calcification measured via BW.

b Corals were already naturally exposed to low pH conditions.

c e.g., lateral and cross-sectional growth, linear extension, skeletal density.

In the present study, the 45Ca and TA anomaly technique for estimating NC and GC, respectively, were compared on long-term acclimated (16 months) Stylophora pistillata microcolonies under ambient and reduced pH in order to test their compatibility in providing consistent calcification estimates even in the context of OA research. Since TA calcification measurements are commonly combined with respiration measurements using closed (sealed) vessels, it was furthermore tested to what degree incubation time affects the stability of carbonate system parameters and the calcification rates, because especially at reduced pH, biological processes might amplify changes in these parameters due to the reduced buffering capacity of seawater.

Materials and Procedures

Coral preparation and acclimation to pH treatment

Stylophora pistillata was collected by scuba diving from a depth of 4–8 m near the Interuniversity Institute for Marine Sciences (IUI) in the Red Sea, Eilat, Israel (29°30′N 34°55′E), in an area supervised by the Israel Nature and Parks Authority. Only a single colony was used for the calcification measurements in order to remove the effect of genetic variability for the tested treatments. The colony was fragmented into 36 fragments and these were suspended on nylon threads in flow-through water tables supplied with natural seawater under a constant temperature of 25 ± 0.5 °C, regulated by an array of 300 W BluClima aquarium heaters (Ferplast Spa, Vicenza, Italy) in an air-conditioned room. This setting allowed the tissue to grow over the exposed skeleton, creating microcolonies (definition sensu Almoghrabi, Allemand & Jaubert, 1993). All pieces were checked to be free of boring organisms. Light was provided by a metal halide lamp (14,000 K, 400 W/D, Osram GmBH, Germany) and photosynthetically active radiation (PAR) measured as ∼170 µmol photons m−2 s−1 (10 h L:14 h D photoperiod; quantum irradiance meter LiCor, Lincoln, Nebraska, USA). Microcolonies were either exposed to present day ambient pHT 8.1 (pH in total scale; 8.08 ± 0.03; pCO2 = 396 µatm) or pHT 7.5 (7.46 ± 0.04; pCO2 = 2, 057 µatm; Table 2), which corresponds to a predicted pCO2 level for 2,300 with Ωarag close to 1 (Caldeira & Wickett, 2005) and was chosen to provoke some degree of coral dissolution. The microcolonies were maintained under these conditions for over 16 months (thus, relative to most other studies acclimation time was very long, cf. Table 1; Table S1). pH was tightly regulated and monitored over the acclimation period (see below). Corals were fed each week with freshly hatched Artemia sp. nauplii and a mixture of crushed fish.

Table 2 Seawater carbonate chemistry in both pH treatments over the 16 months acclimation period.

TA and pH were measured, while the inorganic carbon speciation and aragonite saturation state (Ωarag) were calculated using the software CO2SYS version 1.03 (Lewis, Wallace & Allison, 1998; Pierrot, Lewis & Wallace, 2006). Values are means ± SD (N = 5).

pHT	TA (µeq kg−1)	DIC (µmol kg−1)	pCO2 (µatm)	CO2(aq) (µmol kg−1)	HCO3− (µmol kg−1)	CO32− (µmol kg−1)	Ωarag	
8.08 ± 0.03	2,496 ± 9	2,120 ± 26	396 ± 39	10.9 ± 1.1	1,846 ± 40	263 ± 15	3.96 ± 0.23	
7.46 ± 0.04	2,494 ± 11	2,436 ± 23	2,057 ± 211	56.4 ± 5.8	2,301 ± 23	79 ± 6	1.19 ± 0.10	

Control of seawater pCO2

Seawater was pumped from a depth of 30 m into 1,000 L tanks, where the pH was manipulated to reach a fixed value. The incoming seawater presented a very stable chemistry during the whole experiment period: salinity of 40 ± 0.2‰, pHT of 8.09 ± 0.02, total alkalinity of 2, 505 ± 0 µeq kg−1, as reported by the Israel National Monitoring Program (NMP) of the Gulf of Eilat (NMP, 2016). A pH electrode (S-200C, Sensorex, CA, USA) located in each water table was connected to a pH controller (Aquastar, IKS ComputerSysteme GmbH, Karlsbad, Germany), monitoring the pH (software Timo, Matuta, Germany) and adjusting the bubbling of CO2 (from a CO2 cylinder) accordingly. Well-mixed water from each tank flowed continuously into the corresponding section (150 L) of the water table and pH was monitored daily. Seawater samples from the water tables were taken five times over the acclimation period to monitor TA (TA was stable with 2,494 ± 11 and 2,496 ± 9 µeq kg−1 for pHT 7.5 and 8.1, respectively; Table 2).

Incubation procedure for measurements of calcification

After 16 months of pH acclimation in water tables, microcolonies (2.5–3.0 cm in length, N = 6 per treatment and time point) were incubated in sealed (no gas exchange; no headspace) incubation vessels (40 mL Septa vials with additional paraffin wrap, clear borosilicate glass; 2.5 ×10 cm; Thomas Scientific, Inc, Swedesboro, NJ, USA) containing ∼40 mL filtered seawater (FSW; 0.45 µm) at the two pH levels for 2 h, 4 h, and 6 h. Temperature and light intensity during the incubation were similar to those provided during the preceding 16 months of acclimation (see above), while an orbital shaker provided constant water motion. Applicability of the shaker in contrast to conventionally used stir bars was tested in a preliminary experiment (Fig. S1). Prior to the incubation, the colony surface and attached nylon strings were carefully cleaned of epiphytes and algae. Microcolonies in the incubation vials were hanging without any wall contact by using a T-shaped string (Fig. S2). The TA and 45Ca experiments were carried out over two consecutive days, using the same microcolonies and time of day to avoid effects of diurnal variations in physiology (Edmunds & Spencer-Davies, 1988). After TA measurements, the respective microcolonies were returned to the water table overnight and used for 45Ca measurements the following day. This allowed minimizing physiological differences due to handling or cumulative stress between both measurements. The chosen repeated measures design clearly represents a practical compromise. Although the dual measurement of the 45Ca-uptake into the skeleton and the TA change in the surrounding seawater would have been preferable, the subsequent alkalinity measurement of radioactive seawater with a radioactive contamination of the titrator and the associated equipment was not justifiable for reasons of lab safety (note however that such an approach was taken by Tambutté et al., 1995). While day-to-day variations in calcification cannot be ruled out with certainty, we found no significant statistical effect of this in the complete dataset (i.e., no within subjects effect in separate 2 h, 4 h, 6 h incubations; Table 3).

Table 3 Statistical output.

Repeated measures ANOVA results for calcification rates determined from the TA or 45Ca technique. Asterisks indicate significant results. Note that these rates are derived from fragments of a single biological replicate for the purpose of methodological comparison.

	F-value	p-value	
Complete dataset (2 h, 4 h, 6 h)			
Between subjects			
pH	F1,30 = 4.565	p = 0.0409*	
Incubation time	F1,30 = 1.850	p = 0.1748	
pH × incubation time	F1,30 = 2.581	p = 0.0925	
Within subjects			
Method	F1,30 = 3.117	p = 0.0877	
Method × pH	F1,30 = 1.210	p = 0.2801	
Method × incubation time	F1,30 = 0.690	p = 0.5093	
Method × pH × incubation time	F1,30 = 2.484	p = 0.1004	
2 h dataset			
Between subjects			
pH	F1,10 = 110.748	p < 0.0001*	
Within subjects			
Method	F1,10 = 20.020	p = 0.0012*	
Method × pH	F1,10 = 7.152	p = 0.0233*	

Total alkalinity anomaly technique

Net calcification rates of corals were determined using the TA anomaly technique (Chisholm & Gattuso, 1991). Water samples were collected for TA at the beginning and end of the incubation from each treatment to determine carbonate chemistry and stored until analysis in sealed, bubble-free brown glass vials (borosilicate glass) in the dark at 4 °C. Calcification rates were calculated from the difference between TA measured at the beginning and the end of each incubation period (ΔTA; corrected for blank values from filtered seawater only incubations with N = 3), according to the equation by Schneider & Erez (2006). Rates were normalized to skeleton dry weight: Calcificationμmol CaCO3h−1g−1dry skeleton=ΔTA2×Vvessel−Vcoral×1,000×1.028W×T

ΔTA (in meq kg−1); V vessel is the volume of the experimental vessel (in mL); V coral is the displacement volume of the coral (in mL); 1.028 is the density of seawater in the northern Gulf of Eilat (in g mL−1); W is skeleton dry weight (in g) and T is the incubation duration (in hours).

45Ca technique

Measurements of GC were based on the improved 45Ca protocol (Tambutté et al., 1995). Microcolonies were placed in filtered seawater (FSW; 0.45 µm) with a total activity of 360 kBq (as 45CaCl2, 1958.18 MBq mL−1, Perkin-Elmer Life and Analytical Sciences). Dead microcolonies, killed with 2% formaldehyde, were included in the experiment as a control for isotopic exchange (Al-Horani et al., 2005) and three incubation vessels were left without microcolonies to serve as a seawater-only control. To determine the specific activity, 100 µL aliquots were taken at the beginning and end of each incubation period. Following the labelling period, specimens were immersed in 600 mL FSW for 1 min, and then rinsed (5 ×1 min) with 10 mL of ice-cold glycine-high calcium medium (50 mM CaCl2, 950 mM Glycine, pH 8.2). Labelled specimens were then incubated for 30 min in vessels containing 20 mL of FSW. Water motion was provided by a shaker. Following efflux incubation, microcolony tissue was removed using 2 M NaOH for 20 min at 90 °C. Following tissue hydrolysis, the skeleton was first rinsed with 1 mL NaOH (Houlbrèque, Tambutté & Ferrier-Pagès, 2003), then thoroughly rinsed with FSW, followed by double distilled water. The solution from the first rinse was added to the tissue hydrolysate, while the remaining rinsing solutions were discarded. Finally, skeletons were dried at 70 °C, weighed, and subsequently dissolved in 12 M HCl. Samples (500 µL) of skeleton digest or tissue hydrolysate were added to 10 mL Ultima Gold AB (PerkinElmer) scintillation liquid and measured on a scintillation counter (Tri-carb 1600TR, Packard). Sample count rates (CPM, total counts per minute) were obtained several times over two days and the average difference between readings was 0.8 ± 0.54%. Total calcification was then calculated from the activity recorded in the skeleton digest and seawater control samples and normalized as µmol CaCO3 per skeleton dry weight using the formula: Calcificationμmol CaCO3h−1g−1dry skeleton=Activitysample×1.17ActivityseawaterW×T

where Activitysample is the total counts per minute (CPM) in the dissolved skeleton sample; Activityseawater is the total CPM in 100 µL seawater sample (control); 1.17 is the amount of Ca2+ in 100 µL ambient seawater (in µmol); W is skeleton dry weight (in g) and T is the incubation duration (in hours). The amount of 45Ca uptake by dead specimens (covered with tissue) was subtracted from the amount measured in intact (live) specimens.

Calculation of seawater carbonate system

Value of TA of accurately weighed seawater samples (12.3 g) were measured to the second end point (Almgren, Dyrssen & Fonselius, 1983) using an automatic potentiometric titration (Mettler-Toledo GmbH, DL67 titrator; resolution of burette volume: 1/5,000). The TA was determined in triplicate and computed using the Gran equation (DOE, 1994) with pH values lower than 3.9 for creating the Gran plot. Accuracy of TA analysis was checked against certified seawater reference material prepared by Andrew Dickson (Scripps Institution of Oceanography). The differences between triplicate samples were less than 6 µeq kg−1. The pH electrodes (Mettler-Toledo DG-111–SC; Stockholm, Sweden) were calibrated daily before using the titrator (the manufacturer’s technical specifications are ±0.03 pH for both precision and accuracy). The acid concentration was 0.049 N HCl (JT Baker, Phillipsburg, NJ). In the series of experiments that compared gross and net calcification, a new titrator was utilized: a Metrohm 862 compact Titrosampler that uses at least 35 g seawater samples (autosampler combined with titrator; potentiometric measuring accuracy of ±0.003 with resolution of 0.001; resolution of burette volume: 1/10,000). Hence, experimental samples containing only 40 mL, were diluted by a factor of three (to have enough water for triplicate measurements of TA) and the acid concentration was set to 0.025 M. TA was calculated using the first derivative of the curve for the evaluation of the exact end point. Prior to measurements, water samples were filtered (0.22 µm EMD Millipore Millex sterile syringe filters). Water samples for analysis were stored in darkness at 4 °C in brown glass bottles, filled up to the top with a gas tight screw, and processed within two weeks of collection.

pH measurements were carried out using a CyberScan pH meter (pH/Ion 510 Eutech Instruments with automatic temperature compensation) and CyberScan gel-filled pH combination electrode. Prior to experiments, the pH electrode was calibrated against National Bureau of Standards (NBS) scale buffers of 4.01, 7.00 and 10.00 (Mettler Toledo) at 25 °C and was soaked in seawater for at least 1 h before measurement. The manufacturer’s technical specifications of the pH meter were 0.01 pH for resolution and ±0.01 (standard error) for accuracy.

Components of the carbonate system (pCO2, CO32−, HCO3−, DIC concentrations and Ωarag) were calculated from TA along with pH values, temperature and salinity using the CO2SYS program, version 1.03 (Lewis, Wallace & Allison, 1998; Pierrot, Lewis & Wallace, 2006). The pHNBS were shifted onto the total pH scale (pHT) by subtracting −0.11 (Zeebe & Wolf-Gladrow, 2001), which includes a minor correction for [SO42−] and the stability constant of HSO4− at a salinity of 40.7‰. The thermodynamic carbonate dissociation constants for activity scales (K1 = 5.845 ± 0.008 and K2 = 8.945 ± 0.013) were attained from Mehrbach et al. (1973) and the refit from Dickson & Millero (1987). The input parameters for pressure (10 dbar), total P (0.03 µmol kg−1) and total Si (0.68 µmol kg−1) were obtained from the NMP of the Gulf of Eilat (NMP, 2016).

Data normalization

For accurate determination of TA changes in the incubation volume, coral volume was determined by measuring its displacement weight in seawater following Archimedes’ principle and assuming an approximate density of 1.028 g cm−3 for Red Sea seawater. Normalized calcification rates were obtained by determining skeleton dry weight, using a vibra balance (Shinko Denshi Co., Ltd., Japan; accuracy ∼1 mg).

Statistical analysis

Due to the paired nature of the measurements, the overall dataset was analysed with a repeated measures ANOVA. This way, the within subjects effects allow to test for consistent differences in calcification rates between both estimation methods (TA vs. 45Ca) as a function of incubation time (2 h, 4 h, 6 h), seawater pHT (8.1 vs. 7.5) and the combination of both, whereas the between subject effects test whether overall calcification rates differ between pH and incubation time. All calcification rates were cube root transformed to achieve normality as tested by the Shapiro–Wilk test. Differences between estimation methods due to seawater pH were also tested for the 2 h time point, using the overall fitted model (repeated measures ANOVA as above) as well as a paired t-test for each pH.

The relationship between absolute estimates of calcification (in nmol) between TA and 45Ca was evaluated using geometric regression fitting and by directly comparing 45Ca/TA ratios over a range of calcification values. Geometric regression was preferred over linear regression since no causality can be established between both variables and both have an inherent measurement error. Note that geometric regression equations still depend on the assignment of x and y. To test whether the relationship between both methods is affected by seawater pH, 45Ca/TA-ratios from both treatments after 2 h were tested with a t-test. All statistical analyses were performed in JMP 11.2.1 (SAS Institute, Cary, NC, USA).

For comparison with literature data from Smith & Kinsey (1978) and Tambutté et al. (1995), the software WebPlotDigitizer (Rohatgi, 2015) was employed to extract the data points from the original publication graphs, since the raw data is no longer available (S Smith & E Tambutté, pers. comm., 2015). The extracted data is provided as supplementary information for future reference (Table S2).

Results and Discussion

Effect of incubation time and estimation method on derived calcification rate

During the 16 months of acclimation at both pH treatments all coral microcolonies survived and remained visibly healthy (extended polyps; no bleaching; substantial increase in size). Considering data from all incubation periods, calcification rate estimates from both techniques were not significantly different, nor were there consistent differences between both techniques related to pH or incubation time (Fig. 2 and Table 3). Overall, calcification rates were significantly higher at reduced pH with no significant effect of incubation time (Fig. 2 and Table 3). Since only a single biological replicate was tested, this ecological finding is not of significance and will not be further discussed here.

Figure 2 Calcification rates.

Calcification rates of the same Stylophora pistillata colony at (A) pHT 8.1 and (B) pHT 7.5 derived from different incubation intervals and measurement techniques (45Ca uptake vs. TA) as boxplots with median line (N = 6; technical replicates).

Considering only the calcification rates obtained from 2 h of incubation as physiologically useful data (see later discussion) shows that absolute estimates of calcification differed between both methods depending on pH (Fig. 2 and Table 3). Calcification estimates from the TA anomaly technique were on average 5% and 21% higher than the 45Ca estimate, for ambient and reduced pH, respectively. However, this difference was only statistically significant for the reduced pH treatment (paired t-test, t = 4.486, p = 0.0065∗, N = 6). The implications of this finding, including previously published experimental data, will be discussed in the following paragraph.

Can dissolution simply be derived from the difference between net and gross calcification?

Net calcification rates can be altered as result of changes in dissolution and/or gross calcification (Fig. 1). Distinguishing true changes in gross calcification (as result of biological impairment) from changes in gross dissolution rates is therefore of importance to assess the real impact of pH stress. Paradoxically, the few studies that measure GC and NC that would allow an estimate for dissolution found higher rates for NC in comparison to GC (Rodolfo-Metalpa et al., 2015; Tambutté et al., 1995), with the exception of Smith & Kinsey (1978). Some techniques might not be suitable at all for such calculations, as indicated by a 7- to 11-fold overestimation of calculated GC, when using rates of NC and dissolution from the buoyant weight technique and comparing it with 45Ca GC rates (Rodolfo-Metalpa et al., 2015). In this case, the raw data even suggested that buoyant weight-based dissolution rates in the cold-water coral Desmophyllum dianthus at pH 7.7 were larger than actual 45Ca GC rates, despite measurable NC (as calculated from provided raw data; R Rodolfo-Metalpa, pers. comm., 2015). Correlating blank-corrected calcification values from the TA and 45Ca technique from the 2 h dataset at both pH treatments confirms the strong significant positive correlation between both techniques (Fig. 3A), consistent with previous findings (Smith & Kinsey, 1978; Tambutté et al., 1995). Nevertheless, for fully covered coral colonies (this study; Tambutté et al., 1995), it still suggests that TA tends to overestimate calcification values relative to the 45Ca technique because 45Ca/TA estimate ratios tend to be below 1 (Fig. 3B).

Figure 3 TA vs 45Ca calcification estimates.

Relationship between calcification estimates based on TA titration and 45Ca fixation for Stylophora pistillata (triangles; white and grey this study, black from Tambutté et al., 1995) and Acropora formosa (circle, Smith & Kinsey, 1978). (A) Geometric regressions and pairwise correlations are indicated for all datasets (Table S1) based on Smith & Kinsey (1978) (solid; after 30 min of incubation with A. formosa), Tambutté et al. (1995) (long dash 0.25–3 h incubation with S. pistillata) and this study (short dash after 2 h with S. pistillata pHT 8.1, dash-dot pHT 7.5). (B) Agreement of both methods in relation to absolute calcification value (here based on TA). Data points show ratios of values shown in 3A. Statistical significance of pairwise correlations indicated by asterisks. Double dagger indicates excluded replicate point in the pHT 7.5 data set. Note that it is assumed that Smith & Kinsey (1978) and Tambutté et al. (1995) used blank-corrected calcification estimates.

Previous work by Smith and Roth (in Smith & Kinsey, 1978) incubated tips of Acropora formosa for 30 min in test tubes containing 45Ca-labelled seawater, whereas Tambutté et al. (1995) improved this 45Ca protocol and labelled microcolonies of S. pistillata in seawater for up to 3 h. Both studies confirmed a strong correlation (r = 0.95–0.99) between the TA and 45Ca estimates, but found opposing results with regard to absolute calcification estimates between these two methods. While the slope of the geometric regression was similar between both studies and similar to the obtained values here (Fig. 3A), the regression and its extrapolated intercepts relative to the 1:1 line suggested opposite findings: (1) the presence of 45Ca in the skeleton even without measurable changes in TA (Smith & Kinsey, 1978; Figs. 3A and 3B) and (2) changes in TA without detectable 45Ca incorporation in the skeleton (Tambutté et al., 1995; Figs. 3A and 3B). Physical adsorption of 45Ca to the exposed skeleton surface at the fracture site, rather than biologically mediated incorporation, was suggested as the source for the overestimation by the 45Ca method by Smith & Kinsey (1978). The use of completely tissue-covered microcolonies was intended to minimize this effect in the study of Tambutté et al. (1995) and the resulting negative intercept was interpreted as loss of radioactivity during washing steps or general time lag due to the 45Ca loading of extracellular and tissue compartments. Since both studies chose an overall line-fitting method to compare TA and 45Ca estimates instead of comparing the values directly, it was not apparent that the absolute level of over- or underestimation of either method was not constant, but largely dependent on the absolute amount of deposited CaCO3 (Fig. 3B). This is expected given that the relative overestimation of 45Ca adsorption at exposed skeletal parts, or underestimation due to 45Ca loss, should represent a small, constant value that is related to surface and porosity of the skeleton and is not proportional to the amount of deposited CaCO3. Thus, the influence of such artefacts on the 45Ca/TA relationship becomes less important for larger calcification values (as can be seen in Fig. 3B). We confirmed that the 45Ca isotopic equilibration of dead, covered skeletons (used as blanks) was rather constant after 2 h and 6 h, with 0.14 ± 0.03 and 0.19 ± 0.02 µmol CaCO3 g−1 dry skeleton (mean ± SD; N = 3), respectively. Figure 3B illustrates that for a sufficiently high CaCO3 deposition (>1,500 nmol from TA estimate), the 45Ca/TA ratio approaches a rather constant value in each study with 1.19 ± 0.02 (mean ± SE, N = 3, Smith & Kinsey, 1978), 0.77 ± 0.01 (N = 10; Tambutté et al., 1995), 0.94 ± 0.06 (N = 5; this study pHT 8.1), and 0.85 ± 0.03 (N = 5; this study pHT 7.5). Using only these values and considering only the studies with fully-covered microcolonies, TA estimates are consistently larger than 45Ca estimates. Furthermore, this ratio is not significantly different between the pH treatments in our study (One-Way ANOVA, F1,8 = 1.930, p = 0.2022).

When interpreting the relationship between the 45Ca and TA anomaly technique one must consider two important aspects: (1) the correlational strength, which demonstrates the linear relationship between both methods over a range of values, and (2) the numerical ratio itself, which represents the relative difference in the calcification estimate between both methods and is assumed to provide an estimate for the occurrence of dissolution. For the first aspect, we conclude that both methods correlate well in both pH treatments. For the second point however, we have to conclude that the apparent tendency for GC/NC ratios to be <1 in the relevant studies that used fully covered microcolonies contradicts the generally assumed relationship of NC ≤ GC. This is furthermore problematic since TA estimates in our experiment are factually a measure of GC, because we demonstrated in a separate experiment, that dissolution (if occurring) was not detectable. We tested bare skeletons of similar size in the same vials over a slightly longer period of time (3 h) and even lower pH (pH 8.1, 7.6, and 7.3 with N = 9 each) and found that 25 of the obtained 27 ΔTA values were less than 18 µeq kg−1 and therefore below the recommended detection limit of at least three times the precision of the used titrator (Table S4). Thus, for the case of no detectable dissolution, both methods would represent measures of GC here, but still provide different estimates for gross calcification, especially at reduced pH. Even if dissolution occurred under reduced pH conditions, the 45Ca/TA ratio shifted in the wrong direction, erroneously indicating that NC increases relative to GC. This violates the assumed relationship of NC = GC–GD and the similar results of this study and the study of Tambutté et al. (1995) raises the general question of whether a comparison of these two sensitive methods is really appropriate to provide accurate values for GD. Limited by sensitivity of common methods, assessing gross dissolution and its relation to net calcification simultaneously is clearly a challenge.

Changes in seawater chemistry

Substantial changes in the carbonate chemistry (TA, DIC, pH, pCO2, HCO3−, CO32− and Ωarag) were recorded inside the incubation vessels during the time course of the experiment at both pH treatments (Table 4, Table S3). A general decline in DIC was correlated with an increase in seawater pH and a shift within the carbon equilibrium. For an incubation period beyond two hours, a substantial pH shift (+0.4–0.9 units) with CO2(aq) depletion of more than 70%, as well as considerable changes (≥10%) in seawater DIC were observed irrespective of pH (Table 4).

Table 4 Changes in seawater carbonate chemistry over incubation time.

Values show absolute changes from the initial conditions (mean ± SD, N = 6). Values in brackets indicate relative changes to initial values to account for the shifted carbon equilibrium between both pH treatments. Superscript letters indicate post hoc Tukey HSD test results for detected significant interactions between pH and time in all variables, except alkalinity (Two Way ANOVA results cf. Table S5). Levels not connected by the same letter are significantly different.

	pH 8.1	pH 7.5	
	2 h	4 h	6 h	2 h	4 h	6 h	
Alkalinity (µeq kg−1)	−106 ± 36	−152 ± 48	−198 ± 30	−164 ± 37	−310 ± 115	−345 ± 98	
	(−4%)	(−6%)	(−8%)	(−7%)	(−12%)	(−14%)	
DIC (µmol kg−1)	−215 ± 32	−458 ± 59	−537 ± 82	−277 ± 76	−564 ± 124	−791 ± 103	
	(−10%)A	(−21%)B	(−25%)B	(−11%)A	(−23%)B	(−32%)C	
pH	+0.17 ± 0.04	+0.41 ± 0.06	+0.46 ± 0.10	+0.29 ± 0.11	+0.59 ± 0.09	+0.89 ± 0.06	
	(+2%)A	(+5%)B,C	(+6%)C,D	(+4%)B	(+8%)D	(+12%)E	
CO2(aq) (µmol kg−1)	−4.5 ± 0.7	−8.0 ± 0.6	−8.5 ± 0.8	−34.6 ± 10.9	−52.6 ± 2.9	−60.0 ± 1.2	
	(−42%)A	(−74%)A	(−78%)A	(−53%)B	(−81%)C	(−92%)C	
HCO3− (µmol kg−1)	−276 ± 42	−636 ± 80	−730 ± 134	−293 ± 87	−640 ± 132	−974 ± 109	
	(−15%)A	(−34%)B	(−39%)B	(−13%)A	(−27%)B	(−42%)C	
CO32− (µmol kg−1)	+65 ± 21	+185 ± 33	+201 ± 54	+51 ± 22	+129 ± 31	+244 ± 30	
	(+25%)A	(+70%)B,C	(+76%)C	(+73%)A	(+186%)B	(+352%)C	

At constant temperature and salinity, alterations in carbonate system parameters such as pH and TA are driven mainly by biological activities of the coral holobiont such as photosynthesis, respiration, and calcification (Schulz & Riebesell, 2013). Photosynthesis and calcification decrease the seawater DIC, while respiration and dissolution increase the DIC. However, only calcification and dissolution alter the TA (Chisholm & Gattuso, 1991; Zeebe & Wolf-Gladrow, 2001). The photosynthetic activity of the dinoflagellate symbiont is the main driver for the observed changes in total DIC and DIC speciation (with subsequent pH shift) as CO2 is removed and fixed in biomass at a much higher rate than HCO3−/CO32− incorporation into the skeleton. The stronger changes in carbonate chemistry at reduced pH are likely to be the result of two interacting factors: (1) Potentially altered photosynthetic/respiratory rates at reduced pH (Jury et al., 2013; Page et al., 2016), and (2) a generally lower buffering capacity of CO2-enriched seawater, specifically at pHT 7.5, where seawater reaches its minimum buffering capacity (Delille et al., 2005; Egleston, Sabine & Morel, 2010; Riebesell et al., 2007; Suzuki, 1998). Near pHT 7.5, where DIC∼TA, any changes in DIC and TA will cause a similar magnitude change in [CO2], [H+], and Ω and a sharp change in pH (Egleston, Sabine & Morel, 2010).

Best practice guidelines recommended changes in TA should be at least 3- to 10-fold (Langdon, Gattuso & Andersson, 2010) the analytical precision of the instrument (e.g., ΔTA ∼6 − 20 µmol kg−1 seawater), but these changes along with changes in DIC should not exceed 3% (Schulz et al., 2009) or 10% (Langdon, Gattuso & Andersson, 2010) of the absolute values. Given the observed stronger changes in seawater chemistry at lower pH over time in our experiment, one can see that the trade-off between using a small incubation volume to accurately measure changes in TA versus using a large volume to minimize changes in carbonate chemistry becomes a crucial issue for measurements under OA conditions, especially in closed systems. For calcification measurements with symbiotic corals at reduced pH in closed vessels, incubation time must be minimized as long as the seawater carbonate parameters are not kept constant. In addition, increasing oxygen saturation in closed vessels due to photosynthesis is also an important factor to consider when measuring zooxanthellate corals in the light. In a similar experiment, very high pO2 levels (200–250%) were recorded after 4 h and 6 h of incubation, at both pH treatments and overall O2-productivity significantly declined with prolonged incubation (Fig. S3A, S3B) likely due to a negative feedback of high oxygen levels on photosynthesis and general physiology. Hyperoxic conditions promote the generation and accumulation of reactive oxygen species (Gerschman et al., 1954) with potentially negative effects that interfere with the normal physiological performance and the subsequent estimate of calcification rate.

The problem of fundamental changes in seawater chemistry and their negative feedback for coral physiology can essentially be captured in the relationship between coral biomass, chamber volume, and incubation time. The presented experimental data clearly confirm that these factors define the applicability and limits of closed chamber incubations as highlighted previously, but become especially important in the context of measurements under OA conditions (Chisholm & Gattuso, 1991; Langdon & Atkinson, 2005; Langdon, Gattuso & Andersson, 2010; Schulz et al., 2009). Note that coral size rather than coral surface area is used in the following, since surface area as a size proxy was not always available from the below discussed studies. For comparison, in the present experiment, the ratio between incubation volume (mL) and coral size (cm) was 13 − 16 (the corresponding Vvessel∕Vcoral-ratios for this branching species were 20 to 44) and incubations beyond 2 h already caused large shifts in carbonate parameters and pO2. This should be a general point of consideration, since previous studies that used the same coral species, similar conditions of temperature and light intensity (∼25 °C; ∼150–170 µmol photons m−2 s−1), and used similar (10–17; Furla et al., 2000) or even smaller (6–8; Tambutté et al., 1995) incubation volume/coral size-ratios, incubated corals for up to 3 h (incubation volumes: 7–10 mL), thus most likely experiencing similar shifts in the carbonate species and/or O2 concentration. The other extreme, employing very large incubation volumes relative to coral size, bears the risk of approaching the detection limit of the TA titration method, even with longer incubation times. For example, ΔTA over 2 h approached less than three times the analytical precision for most of the treatments for a volume/size ratio of 225 in the study of Takahashi & Kurihara (2013). It should be favourably noted that the studies that experienced this issue and used alkalinity changes as low as 3 µmol kg−1 to calculate calcification rates belong to the few publications that explicitly report the measured changes in the carbonic acid system in the incubation vessels (Hossain & Ohde, 2006; Ohde & Hossain, 2004; Takahashi & Kurihara, 2013). Our experimental data reaffirm the necessity of reporting changes in the carbonic-acid system parameters not only for pre-experimental pH acclimation, but also for actual incubation experiments that yield calcification data, following previously suggested best practice guidelines (Andersson & Mackenzie, 2012; Langdon, Gattuso & Andersson, 2010). The need for a valid description of the whole carbonate system is further emphasised by the study of Hoppe et al. (2012) on the uncertainties in the calculated carbonate chemistry when using only two measured parameters.

In order to avoid substantial changes in the seawater chemistry in incubation vessels, it might be useful to estimate the maximum permissible coral size based on known estimates of the species-specific calcification rate, the known incubation volume and by applying the 3–10% rule for the permitted absolute TA change. For our case, given a Red Sea seawater TA of ∼2,500 µeq/kg, TA must not change by more than ∼75 µmol L−1 (3%-rule) over the total incubation time, corresponding to ∼3,000 nmol in a 40 mL chamber (note that the real incubation volume is actually Vchamber–Vcoral). For an expected calcification rate of ∼250 nmol CaCO3 cm−2 h−1 for a Northern Red Sea Stylophora pistillata (e.g., Krueger et al., 2017), the maximal permitted coral surface area of the fragment used for a 1 h incubation must be smaller than 12 cm2 (3,000 nmol/250 nmol) or 6 cm2 for a 2 h incubation. Considering the varying performance in calcification, photosynthesis, and respiration between coral species and environmental conditions (e.g., pH, light, temperature), preliminary experiments might also benefit in determining minimal vessel volumes, depending on coral size and respective incubation time in closed systems.

Conclusion

Stating that reduced pH decreases coral calcification based on measurements of net calcification alone fails to identify whether there is a reduced biological capability to form a new skeleton or whether skeletal dissolution outweighs the biologically-mediated deposition (see Fig. 1). We demonstrated the significance of potential changes in TA and DIC speciation as a function of incubation time when using photosynthetically active corals for two common techniques that provide estimates for NC and GC. Given the remaining uncertainties about deriving gross dissolution from different NC-GC comparisons (BW/TA vs. 45Ca), further experiments are clearly required to provide an explicit value for gross dissolution of skeletal material in living corals. One experimental approach that was previously suggested, but to our knowledge has not yet been experimentally tested, is to incubate a coral in 45Ca-spiked seawater for a few hours/days in order to sufficiently label the skeleton so that a subsequent incubation in seawater would allow the direct assessment of 45Ca-dissolution from the skeleton (after correcting for the isotopic equilibration) (Langdon, Gattuso & Andersson, 2010). Simultaneous incubation with a different detectable calcium isotope or a dye such as alizarin as employed by Lamberts (1974) would provide an estimate for gross calcification. This dual approach could allow the simultaneous and direct assessment of rates for GC and dissolution on the same coral by quantifying Ca influx and efflux simultaneously. The importance of accurately measuring these processes, while considering the different factors that can affect the physiological response during incubation, emphasises the need for additional comparative studies to test the compatibility and accuracy of calcification estimates based on the TA and 45Ca techniques in the context of OA research.

Supplemental Information

Figure S1 Calcification rates in shaken vs. stirred 40 mL incubation chambers

Calcification rates of five Stylophora pistillata microcolonies under ambient pH conditions as measured by the total alkalinity technique. Comparison of calcification rates derived from stirring (magnetic stirrer) or shaking (water bath) 40 mL incubation chambers over 1 h of incubation. Both methods are widely used in the literature for generating water motion inside an incubation chamber. Experiments were conducted over two consecutive days using the same fragments and results tested as pairwise t-test.

Click here for additional data file.

Figure S2 Incubation vials used for the TA and 45Ca measurements

Microcolonies were fully covered by tissue and were suspended on nylon threads. Note that corals did not touch the wall of the vials.

Click here for additional data file.

Figure S3 Oxygen supersaturation and reduced productivity after prolonged incubation

(A) Representative graph for general oxygen saturation levels in 40 mL incubation vials over time. 100% saturation corresponds to an oxygen content of 6.581 mg L−1 for the used seawater. (B) Corresponding coral O2-production rates for the three phases of the incubation period at ambient and reduced pH. Means ±SD, N = 3. Two-Way ANOVA; effect of incubation interval F2,16 = 19.918, p < 0.0001; effect of pH F1,17 = 6.035, p = 0.0277; no interaction.

Click here for additional data file.

Table S1 Summary of the available experimental data examining the effect of ocean acidification conditions on coral calcification

The table contains information on tropical reef-building corals only. Please note that for some studies level of response to seawater acidification was affected by an additional environmental factor such as nutrients, light intensity, etc. (detailed in ‘Notes’ column). Incubation time and incubation volume are only relevant for experiments that the TA method was used to derive net calcification rates. 0 value, calcification rates were not significantly different from ambient pH; TA, total alkalinity; BW, buoyant weight.

Click here for additional data file.

Table S2 Raw data used to create Fig. 3 of this study

Data were extracted from Fig. 2 in Smith & Kinsey (1978) and Fig. 5 in Tambutté et al. (1995), using the software WebPlotDigitizer (Rohatgi, 2015). Values from Smith & Kinsey (1978) were converted using a molecular weight of 100.0869 g/mol for CaCO3.

Click here for additional data file.

Table S3 Raw data of seawater chemistry of incubation vessels

Seawater carbonate chemistry in each of the incubation vessels as recorded in the alkalinity experiment. TA and pH were measured while all other parameters were calculated using the CO2SYS program (Lewis, Wallace & Allison, 1998).

Click here for additional data file.

Table S4 Raw data of skeletal dissolution rates

Rates of alkalinity change of bare skeletons in different seawater pH treatments after 3 h of incubation. Chambers were identical to the ones used in the described experiments. The precision of the used titrator was less than 6 µeq kg−1, thus usable ΔTA values have to be at least 18 µeq kg−1.

Click here for additional data file.

Table S5 Statistical output on seawater chemistry changes

Two-Way ANOVA results on effects of pH (pHT 8.1 vs. 7.5) and incubation time (time period 2 h, 4 h, 6 h) on changes in variables related to seawater chemistry shown in Table 4. Asterisks indicate statistical significance with N = 6.

Click here for additional data file.

The authors thank J Erez and K Schneider for comments and ideas throughout the experiments, R Rodolfo-Metalpa for critical discussion on certain aspects of the manuscript, S Krief and L Hazanov for technical assistance and the staff of the Interuniversity Institute for Marine Science in Eilat.

Additional Information and Declarations

Competing Interests

Author Contributions

Data Availability

The authors declare there are no competing interests.

Stephanie Cohen conceived and designed the experiments, performed the experiments, analyzed the data, wrote the paper, prepared figures and/or tables, reviewed drafts of the paper.

Thomas Krueger analyzed the data, wrote the paper, prepared figures and/or tables, reviewed drafts of the paper.

Maoz Fine conceived and designed the experiments, performed the experiments, contributed reagents/materials/analysis tools, wrote the paper, reviewed drafts of the paper.

The following information was supplied regarding data availability:

The raw data has been supplied as Supplemental Files.

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
