# Peer review of "Measuring coral calcification under ocean acidification: methodological considerations for the 45Ca-uptake and total alkalinity anomaly technique"

_PeerJ, doi:10.7717/peerj.3749_

## Round 0.1 · original submission · Major Revisions

Dear Author,

Whilst PeerJ doesn't make judgements based on impact, novelty or interest I think the reviewers have raised sufficient concerns on the methodology and results to feel the paper needs more time to rework the data into a publishable form. For this reason I have recommend major revision in this instance in order to give you time to address all the concerns raised by the 3 reviewers. In particular I strongly recommend to look at the methodological issues raised by Reviewer 3 together with the concern about the accuracy of the data.

best regards

Federica Ragazzola

Reviewer 1 ·

Basic reporting

The organization of the introduction includes topics relevant for the research questions investigated in this study. The authors compared the rates of gross and net calcification estimated from two contrasting methodologies (TA anomaly and Ca-uptake) under ambient and low pH. Additionally, they wanted to examine how much carbonate chemistry parameters and consequently calcification rates change with varying incubation times. The introduction and methods are well-written (Fig 1 shows a nice conceptual diagram) but the discussion is very difficult to follow due to the organization. It could be improved by first presenting the results and discussion from the calcification rate estimations (research question 1) and then introducing the caveats of length of experiment on these rates and carbonate chemistry (research question 2). Specific comments are provided in the “general comments” section. However, that being said, the science is sound and addresses important considerations for conducting physiology, particularly calcification, experiments under OA in a laboratory setting.

The authors should carefully review their manuscript as there are several places where unneeded parentheses are present and superscripts are not formatted (e.g. in equations for calcification rates). Supplemental figures were referred to in the text but not provided and the caption for Figure 3 does not explain part B. Also, there is no error or variability shown in Table 3 which summarizes the carbonate chemistry of the water tables.

Experimental design

The authors present original research that addresses their main research question of comparing two techniques to measure gross and net calcification under ambient and acidified conditions. They build on previous research by introducing the OA component and perspective. However, the methods are not completely clear, particularly for the equations they used to calculate NC.

Please keep significant figures consistent throughout the methods and results. There are times when pH is reported to the hundredth decimal place and other times when it is reported to the thousandth decimal place. Same for TA.

Line 119-120: pH was maintained at a stable level? Is this representative of what the coral colony experienced in the field? Also, the variability for the low pH treatment seems quite large compared to the ambient treatment – possible reasons?

Line 136: It’s unclear what is meant by “pH system” as this could refer to the tank or water tables.

Lines 146-149: Where are Fig S1 and S2? This would be helpful for the reader in order to understand the experimental methods and design.

Line 166: Were the samples not treated with mercuric chloride to halt any biological activity?

Line 171 (equation): what is the 1000? If this is a conversion for kg to g, the equation should be divided by 1000 rather than multiplied by 1000.

Line 180-181: switch these sentences to create better flow

Line 209: should be “Scripps Institution of Oceanography”

Validity of the findings

The findings of this study are quite interesting and the data are robust and statistically sound. However, as mentioned earlier, I would suggest re-organizing the results/discussion to address your first research question (compare estimates of NC and GC by two techniques under ambient and reduced pH) before discussing the carbonate chemistry stability over various incubation times. It would greatly improve the reader’s understanding of the results and how biological processes relate to carbonate chemistry changes over time and changing ocean conditions (ambient vs OA).

The authors found surprising calcification responses to OA (higher calcification under OA) both in the overall data and when comparing rates after 2 hours of incubation. These results, while contradictory to most of the published literature, are quite exciting and should be briefly discussed. It’s very interesting that this response occurs despite all of the coral microcolonies containing the same genetic code. Could these results indicate some potential for upregulation of some calcification genes or simply faster growth in response to OA? (This is speculation, and should be indicated as such, but could lead to further studies.) There is a nice discussion of the caveats for each method of measuring calcification rates which could influence these measurements but the results are interesting nonetheless!

Line 284-285: buffering capacity can certainly explain a larger change in DIC/pH but also changes to rates of photosynthesis and respiration which have been shown to be altered when corals experience OA (for example, see discussion in Jury et al 2013 and Page et al 2016). I would suspect that for this system, the observed differences in magnitude of changes are due to both buffering capacity changes and alterations to coral physiology.

Line 370: Figure 3B actually shows ratios both above and below 1 when TA is lower than 2000 not just below 1.

Line 397: Is the TA reported in figure 3b the change owing to NC or simply the TA measured in the incubation chamber? (From the text, it sounds like Fig 3b shows the change in TA owing to net calcification since higher TA indicated higher calcification. However, this plot is not described in Figure 3.) The interpretation of this axis will influence the conclusions from this analyses.

Lines 413-415: It is unclear how gross calcification and absolute calcification are defined in the context of this paper.

Additional comments

Line 20: change “calcification values” to “calcification rates”
Line 41: change “pCO2” to “CO2”

Line 45: may affect many marine organisms. Please provide literature examples (such as Ries et al, Kroeker et al)

Line 52: change “overall dissolution” to “overall calcium carbonate dissolution”

Line 54-55: I would not agree with this statement that “projections predict that many of them will even go from net accretion of reef skeleton to net dissolution.” A few studies do bring up concerns that reefs may undergo shifts from net accretion to net dissolution and erosion. However, it is still uncertain when and how quickly such transitions will occur. The authors should provide more evidence/citations or tone down the statement.

Line 72: the equation should be represented as NC=GC-GD in order to match with the cited studies

Line 80: please cite Jokiel et al. 1978 for the buoyant weight technique

·

Basic reporting

The language used throughout is generally clear. However, there are some areas where I have found errors or suggested alternative ways. Please see attached annotated pdf.

In general, the references in good and a comprehensive backround is given. However, I found some things that need references. As well, the author should make sure all references are in chronological order and seperated with a semicolon, not a comma. Please see attached.

Thank you for providing the raw data. I have only found one issue with Table 1 that required clarification. See attached.

All results are within the scope of the mansucript and relevant to the hypothesis tested.

Experimental design

The methods are soundly stated. I am wondering if the authors can elaborate on why such a low pH was chosen and not something more ecologically relevant for near future? Page 7, line 120.

While I am familiar with alkalinity technique, I have never performed calcification estimated via 45Ca technique. Therefore my comments are minor if not any for that section and I would suggest to the editor that the other reviewer have initimate knowledge of this technique to comment.

I have few comments on this section in the attached pdf.

Validity of the findings

The conclusions and data presented are well dicussed in the wide literature. This study critically assesses two methods for comparing calcification rates that adds to the wide literature on the need for more comparable estimated between methodologies. It is important work that fits well within the standards of PeerJ.

Reviewer 3 ·

Basic reporting

No problems with basic reporting

Experimental design

The study is meant to answer the question, does a reduction in pH reduce gross calcification, does it increase dissolution or is it a combination of the two? They also make some best practices recommendations about picking the size of the coral and the volume of water incubated so that the changes in water chemistry during the incubation are small relative to the differences between the treatments.

Taking the question, can we measure dissolution as the differnce between GC and NC first? After doing their experiments they are forced to conclude that they can't tell from their data. The problem is two-fold. First, the assumption that GC based Ca-45 uptake > NC based on TA change is violated. Often they find that net calcification exceeds gross calcification which should be impossible.(Fig 3B). Second, the errors on the measurements of GC and NC are large (Fig 2). If dissolution is some fraction of net calcification they are not going to be able to quantify a difference that is significantly different from zero. An error analysis of the GC-NC method would be helpful.

As they mention the observation NC based on TA > GC based Ca-45 fixation has been observed before. What to make of this? The use of Ca-45 to measure GC may have some problems. If you look at Fig 3A you can see that in some past work data sets have a positive intercept if Ca-45 fixation is plotted against TA based calcification. This implies Ca-45 fixed into skeleton when no net calcification is happening which seem unlikely. Put another way there may be a Ca-45 blank issue that hasn't adequately been corrected. In other data sets the intercept is negative suggesting in this data set the Ca-45 blank correction was overestimated. Of course we can't rule out that there are not issues with the TA measurements of calcification also. Regardless, it raises concerns about the accuracy of the data at a level that may preclude trying to measuring dissolution from the difference of the two measurements.

Never addressed in this manuscript is the question, did the treatment (pH 7.5) have any significant effect on the calcification of S. pistillata as measured by either method? Looking at Fig 2 it looks to me like the rates at pH 7.5 are a little higher than at pH 8.1 but perhaps not significantly higher. As far as I can tell the authors don't address this interesting result. Instead they focus on the difference between the two methods and whether that difference is affected by pH. Here they do report some significant pH effects. But in the one case where there is a difference ie after 2 h TA is greater than Ca-45 which does not make sense.

I think the conclusion is that one or both methods have problems in this study and that the authors have not yet sorted them out to the point that they have a publishable result to report.

There are methodological issues that may contribute to their problems and these go to best practices.

1) They do not mention feeding the corals. In my experience over weeks to months unfed corals loose tissue, the photosynthetic efficiency of the symbionts declines and growth rate measured by BW declines. It possible that after 18 months in culture these corals where not particularly healthy. It is best practice to measure the growth rate of corals before the start of an experiment and express the rates on a gram per cm2 per day basis so that the rates can be compare unambiguously with other studies so the investigator can know that the corals are healthy and growing at near natural rates.

2) Corals like water motion. I am not sure that the corals in this experiment sealed in 40 ml vials were getting enough water motion. Agitating the vials may not have imparted much relative motion between the corals and the water if there was no head space.

3) In these experiments the change in carbonate chemistry during the incubation was large. You want to keep the pH or omega change during the incubation small relative to the difference between treatments. The pH difference between the treatments was 0.6 units. Looking at Table 3 it can been seen that the pH increased over the course of the incubations by >0.4 units in all but the 2 h incubations. The authors rightly excluded all of their data from the 4 and 6 h incubations but that didn't leave them much data for their statistical analysis. As the authors mention it is best practice to limit the TA changes to 3%. This will keep the pH change to 0.1 units in the control experiments. In the low pH treatments the pH increase will be 0.2 units and the increase in omega will be 43% and the reduction in CO2 -40%. Clearly you wouldn't want the changes to be much larger. Minimizing the chemistry changes during an incubation is an important part of doing good OA experiments. The authors rightly stress this point. However, their recommendation of using the length of the coral seems inadequate to me. There is quite alot of calcification data available for S. pistillata and synthesizing that data I find that the calcification rate ranges from 600-1800 nmol CaCO3/cm2/h. Applying the 3% rule you can calculate that TA should not be allowed to change by more than 75 umol/L or 3 umol or 3000 nmol in a 40 ml vial. Dividing 3000 by 1800 and by the incubation time in hours gives the size of the coral in terms of surface area that should not be exceeded. For a 2 h incubation the surface area of the coral should not exceed 2.5 cm2.

Validity of the findings

The authors conclude that the TA-based calcification rates in the pH7.5 treatment are erroneously higher than the Ca-45 based rates. I agree that the two measurements are different but I do not see a sound reason for concluding which one is more right. Both of them are higher than the controls which seems far more significant to me. I don't trust either of the treatment rates. Were the corals in good condition? Were they stressed by the tiny vials and lack of water motion?

The authors mention an alternative method for measuring dissolution but it wasn't tested in this study so you don't know if the method is good or not.

---

## Round 0.2 · accepted · Accept

Dear Authors,

The reviewer who agreed to re-read the manuscript is happy with the new version.

Best regards
Federica

Reviewer 1 ·

Basic reporting

Be sure to format the super- and subscripts within the equations provided in the methods.

Experimental design

no comment. (The authors sufficiently addressed the reviewer's comments.)

Validity of the findings

no comment

Additional comments

no comment